# Understanding Video from Encoded Bytes

## Abstract

We present an approach to understand video from encoded bytes, e.g., mp4s. These compressed videos are 99% smaller than the RGB pixel representations which are currently commonly used for video understanding. Encoded videos are able to compress the pixels by taking advantage of the redundant information across the frames using special encoding, such as key frames and motion residuals to handle this. However, standard video understanding models do not take advantage of this significant compression already available for each video, and instead either heavily subsample the frames or only work on short segments of the video. Here, we present an approach to understanding video from encoded bytes directly. We note that simply applying existing models, e.g., Transformers or State-Space models, to video byte sequences does not work, both due to difficulty in handling very long video byte sequences and easy overfitting. To address these challenges, we design a State-Space model with sequence parallelism to handle very long byte sequences, reaching **15 million tokens** in training, and essentially unlimited tokens in inference. We also propose a multilevel SSM activation fusion that reduces sequence length, which we find also benefits video understanding. We evaluate on common video understanding and natural extension to video + audio understanding tasks and demonstrate competitive performance, illustrating, for the first time, the feasibility of learning from compressed video byte representations.

## 1 Introduction

Video understanding is an important problem in computer vision. Unlike images, videos contain many frames. In traditional settings, these frames are treated as a sequence of images (Simonyan & Zisserman, 2014; Carreira & Zisserman, 2017; Yue-Hei Ng et al., 2015; Tran et al., 2015), which greatly increases the compute costs and memory requirements and makes it hard to scale to longer and longer videos. Here, we propose an alternative approach to instead understand videos as encoded bytes, e.g., mp4 byte streams. The main advantage is the significant memory savings in processing compressed video, since the compression codecs take advantage of redundant pixels in consecutive frames, they are greatly able to reduce the size of a video. We also note that the video byte streams are naturally suited for sequential models, as they were designed for video streaming and playback, where the decoders reconstruct the sequential frames from the bytes, and further, byte streams do not contain strong inductive biases, and so do not require operations like convolution as in ViTs (Dosovitskiy et al., 2020). While byte-based representations are highly compressed, both pixel and byte based representations contain the same information, and both have a complex, inconsistent, non-linear mapping between the inputs and semantic understanding, thus it is theoretically possible to learn any video understanding task from either input.

As a motivation, using the standard video representation of float16 $[F \times H \times W \times 3]$, and assuming 30fps and 480 resolution, one would need roughly 40 megabytes of memory per second of video for storing just the pixels (not including the model weights or intermediate activations). For a 10 minute video at 30 FPS and 480 resolution, this would use roughly 25GB of memory just for the pixel inputs. However, that same video encoded with a bitrate of 2Mbps (YouTube's recommended compression rate for 480p videos), would use only about 150MB for the same 10-minute video, which, compared to 25GB, yields a significant reduction in filesize, i.e., about 99% smaller. If we are able to perform video understanding on compressed byte inputs, this will allow for great compute savings and enable scaling to long videos.

| Sequence Length | ActivityNet-QA Acc (%) |
|---|---|
| 250,000 | 42.5 |
| 2 Million | 54.2 |
| 15 Million | **56.3** |

| Sequence Length | ActivityNet-QA Acc (%) |
|---|---|
| 15 Million | 56.3 |
| 20 Million | **57.1** |

Table 1: Our approach excels at handling incredibly long byte sequences from raw compressed video formats, supporting up to **15 Million byte tokens** during training, leading to significant performance improvements.

Table 2: Testing on longer sequence lengths (**20 Million bytes, which corresponds to ∼13 minutes of video**) than training (15M bytes) yields further gains, and can theoretically support unlimited tokens during inference.

Further, existing specialized video models either focus on modeling sparse temporal relationships on image-based features (e.g., (Yue-Hei Ng et al., 2015; Piergiovanni et al., 2017; Chen et al., 2023; Lin et al., 2025)) or use tube-based features on short segments of video (Arnab et al., 2021; Piergiovanni et al., 2023) or apply segment-based pooling (Shou et al., 2016). These prior works perform well on video, but are essentially leveraging existing image-based models for short or sparsely sampled video segments. Instead here, based on the observation that compressed videos contain all the information needed to reconstruct the pixels, but with much less redundant information, we design an approach to directly learn specifically from video bytes. We also note that most video training datasets contain very short clips, and datasets focused on long videos (LVBench (Wang et al., 2024a), MLVU (Zhou et al., 2024), VUE-TR (Team et al., 2025b), Neptune (Nagrani et al., 2024), etc.) are focused on the evaluation of models, since training on long videos is still a challenging task. Even recent works, such as InternVideo2 (Wang et al., 2024b) and VIDI (Team et al., 2025b) only train on short segments with 8 frames and 1 fps up to 120 frames, respectively, and Hour-LLaVA (Lin et al., 2025) very sparsely samples video tokens from a frozen image encoder.

Training on video bytes allows using all the frames with much less memory, however, as we show in the experiments, this is a non-trivial task due to a few issues. First, encoded video bytes are very long sequences. Instead of a video being a $[F \times H \times W \times 3]$ that can be further compressed with spatial and temporal pooling, we have an input of $[L \times D]$, where $L$ is the sequences length and $D$ is the embedding size of the model. For video bytes, this results in sequences with millions of tokens, which is extremely long even compared to modern LLMs (e.g., LLAMA 3.1 (Grattafiori et al., 2024) supports 128,000 tokens in inference, Gemma (Team et al., 2025a) was trained with an 8k sequence length). This presents a real problem, as this becomes a long-sequence length learning problem. Second, we find that Transformers are not the best suited model for this task, due to the long sequences and poor scaling of self-attention. Finally, understanding the encoded representation is far more challenging than understanding pixels, as the representation is much more compressed. Thus, taking an existing LLM model and directly training it on encoded video bytes does not perform well at all. We propose a method to learn from encoded video bytes that works on very long sequences, which are of different structure than text inputs, and we find a multilevel, sequential modeling of video bytes works for this. In this work, we make several key contributions that enable the understanding of videos from bytes:

- We present the first approach to understand video from raw, encoded bytes, circumventing the decoding process which increases the video volume processed by over 100x, and leveraging the highly compressed video inputs which are ubiquitous video representations.

- We propose an efficient parallelization and gradient accumulation method, with a novel correction and propagation of the state, that enables training on extremely long bytes sequences, e.g., **15 million** ( Tables 1, 2), and theoretically unlimited inference length.

- We present a 'multilevel' SSM which efficiently accumulates SSM activations and, together with the sequence parallelism, enables scaling to very long sequences during both training and inference. This model greatly outperforms standard SSMs and Transformers when applied to encoded video bytes (Table 9).

- We find that data augmentation and pre-training on video bytes is extremely important, and without such training, the model greatly overfits and generalizes poorly. We note that here we do not use any prior knowledge about the structure of the bytes, e.g., information about the codec, which could be further leveraged in future work.

## 2 LEARNING FROM RAW VIDEO BYTES

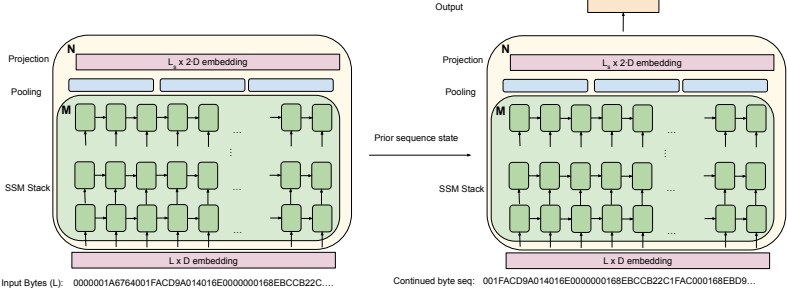

Figure 1: Model outline. The bytes are embedded, then processed by $M$ Mamba SSM layers in a multilevel module (Sec. 3) with pooling between layers, which is repeated $N$ times. This is a sequentially parallel model (Sec. 3.1) which accumulates and corrects the state, allowing for efficient paralelization and extension to long sequences.

Our approach takes a sequence of raw video bytes as input, rather than pixels as in most prior works. For this work, we use the standard h.264 codec (Wiegand et al., 2003) and mp4 container (mp4, 2020) to obtain the video bytes. One observation is that these codecs are designed for streaming video. Importantly, this means that when decoding and recon-

structing the pixels, the decoder algorithm does it byte-by-byte, or based on short segments of bytes. This means that bytes that are far apart, e.g., the 10th and 10,000th byte do not really depend on each other to reconstruct the pixels, since they are compressed with streaming in mind. As a result the standard global self-attention in Transformer models is not really needed to understand video bytes. This is a unique characteristic of compressed video bytes, which is not present in text sequences and language modeling. However, to understand a video, i.e., for classification or question answering purposes, the model does need to be able to understand the whole sequence. To address this, we develop techniques which (a) work with extremely long sequences, (b) use sequential modeling to better model streaming video bytes, rather than self-attention as Transformers use. This allows understanding both low-level local features as well as high level details of the full video. This model is shown in Fig. 1 and described further in Sec. 3. Another issue is that the model tends to overfit and not generalize well when given bytes as input, which we present a solution to in Sec. 3.2.

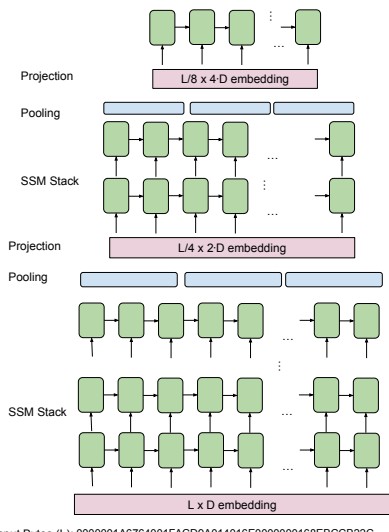

Figure 2: The multilevel SSM. The input is processed sequentially by SSM layers. The pooling layers reduce the sequence length; projection layers increase the dimensionality. This process is repeated multiple times, forming the multilevel SSM. This is further parallelized in order to handle long sequences (Sec. 3.1)

On the other hand, learning from raw bytes presents opportunities for learning from much more economical and efficient compressed video formats which are already readily available, as videos are stored in these formats. Additionally, the approach directly transfers to audio+video inputs, as shown in the experiments. We further save compute in the input pipeline, as we do not need to decode the bytes into pixels , though for larger models this is not the bottleneck, as the compute time of the model is the primary bottleneck.

## 3 PROPOSED APPROACH - PARALLELIZED MULTILEVEL SSM

To learn from very long sequences, we first design a multilevel SSM module, which has a better handle over the sequence information. Recent SSM models (Gu et al., 2021; Gu & Dao, 2023; Gu et al., 2022), which scale linearly and are more suitable for long sequences, are still not able to fully address some main challenges, such as long-range re-

call, recency bias, global sequence understanding, and still have some inefficiencies, e.g., Video-Mamba (Li et al., 2024) trained on only 8 frames and evaluated on up to 64 frames. While some hierarchical SSMs have been proposed before (e.g., (Bhirangi et al., 2024)), we present a different approach which preserves better the sequence signal. We choose to build upon an SSM for a few reasons. First the SSM (Gu et al., 2021) does not use self-attention, thus it scales linearly to long sequences. Second, the SSM processes the bytes sequentially as they are input. Since video codecs like mp4s were designed for streaming videos, an SSM is naturally applicable to this input, processing the bytes and updating the state in the order the bytes are input. Specifically, we use the Mamba (Gu & Dao, 2023) architecture as the base model, given its strong performance among existing SSMs, hardware efficiency, and ability to vary the representation with time (Gu et al., 2021; 2022; Gu & Dao, 2023; Chen et al., 2024). An overview of the multilevel SSM is shown in Figure 2. Given an input sequence, $S$, that consists of $L$ bytes, we first embed the bytes as $D$-dimensional vectors, which are input to Mamba. We then apply $M$ standard Mamba layers. After this, we pool the bytes which decreases the sequences from length $L$ to length $\frac{L}{L_s}$, e.g, $L_s = 2$ to reduce it be half. We explore different forms of this pooling. We repeat this stack of $M$ Mamba layers, followed by a pooling layer $N$ times, forming the multilevel SSM. Finally, we average pool over the remaining tokens and a fully connected layer for classification tasks. Compared to a prior hierarchical SSM (Bhirangi et al., 2024), the key differences is that our SSM is over the whole input sequence, rather over segments and we stack many levels (4 in our experiments) of the pooling, rather than just two levels. Since SSMs scale linearly with sequence length, there is no benefit to splitting the sequence into segments when running the SSM, and applying the SSM to the whole sequence allows the model to have knowledge of the whole sequence through the SSM state.

For the embedding, as there are 256 bytes, we use a vocabulary size of 256 tokens with an embedding dimension ($D$) of 256. We explore the embedding dimension in the ablations. We note that this embedding dimension increases, more memory is used. Despite the fact that using 256 dimensions has more expressiveness than the original 256 bytes, we found using fewer dimensions greatly reduced performance, and going above 256 did slightly improve performance, but also increases memory usage. To address this, we make a few important steps: 1) make the input to the embedding a uint8 type, rather than an int32 as is standard in LLMs (due to their larger vocabulary size). This saves $\sim$75% of memory by needing only 8 bits per token; 2) we use 16-bit precision on the embeddings themselves, finding no difference to 32-bit precision, but saving memory. We evaluate attention, averaging and concatenation as pooling methods (see the appendix for details Sec. A.1).

## 3.1 HANDLING LONG SEQUENCES WITH PARALLELISM

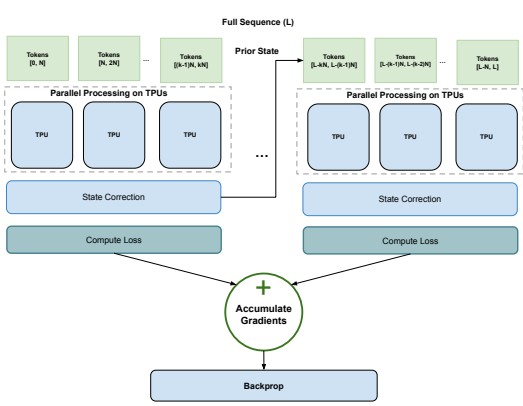

Figure 3: Using sequence parallelism and gradient accumulation to train on sequence lengths of up to 15 million. A key benefit of the SSM is parallel computation of the states, plus a cheap correction calculation (Sec. 3.1), making it easy to scale and suited for streaming video byte data.

Due to the very long sequences that video bytes have, we propose a sequence parallelism technique, utilizing multilevel SSM states. We observe that the SSM outputs can be computed for any part of the sequence without having access to the prior state, then propose a simple update applied when given the state. This allows us to parallelize by placing subsequences on different devices and computing the SSM on each device, then applying an update on the outputs, once each device is done computing. This allows cheap parallelism along the sequence axis, which allows us to scale more efficiently to long sequences. We find that sequence parallelism is around 2x faster than model parallelism for this SSM. Namely, we compute and store the hidden states for each device as if the sequence starts with hidden state set to zero. Consider a signal with $L$ tokens $x = [x_1, x_2, ..., x_L]$ and $V$ devices, we divide the signal into $V$ subsequences each of length $K = \frac{L}{V}$ and apply SSM on each device independently, i.e., the initial hidden state is set to zero for every device. The

final hidden states $h^{(i)}$ (Eq. 1) and outputs $y^{(i)}$ of the SSM (Eq. 2) applied to each of the $V$ subsequences is as given below. Given $A, B, C$ are the parameters of the SSM and $h$ is the state:

$$
\begin{aligned}
h^{(1)} &= Bx_K + ABx_{K-1} + \cdots + A^{K-1}Bx_1 \\
h^{(2)} &= Bx_{2K} + ABx_{2K-1} + \cdots + A^{K-1}Bx_{K+1} \\
h^{(V)} &= Bx_L + ABx_{K-1} + \cdots + A^{K-1}Bx_{(V-1)K+1}
\end{aligned}
\tag{1}
$$

$$
\begin{aligned}
y^{(1)} &= [CBx_1, CBx_2 + CABx_1, \ldots, CBx_K + CABx_{K-1} + \cdots + CA^{K-1}Bx_1] \\
y^{(2)} &= [CBx_{K+1}, CBx_{K+2} + CABx_{K+1}, \ldots, CBx_{2K} + CABx_{2K-1} + \cdots + CA^{K-1}Bx_{K+1}] \\
y^{(V)} &= [CBx_{(V-1)K+1}, \ldots, CBx_{VK} + CABx_{VK-1} + \cdots + CA^{K-1}Bx_{(V-1)K+1}]
\end{aligned}
\tag{2}
$$

The above hidden states are computed in parallel. As observed, they lack the contribution of state from the previous chunks, which are computed on other devices. Thus, the final hidden state from previous sequence shard is added to correct for the missing factor. The equations for corrected hidden states are given in Eq. 3.

$$
\begin{aligned}
h^{(1)}_{corrected} &= h^{(1)} \\
h^{(2)}_{corrected} &= A^K h^{(1)} + h^{(2)} \\
h^{(V)}_{corrected} &= A^K h^{(V-1)}_{corrected} + h^{(V)}
\end{aligned}
\tag{3}
$$

Here $h_0$ is the hidden state to the entire sequence before splitting into subsequences. The equations for the output after the inclusion of correction factor to the hidden state is as given below (Eq. 4):

$$
\begin{aligned}
y^{(1)}_{corrected}[i] &= y^{(1)}[i] \\
y^{(2)}_{corrected}[i] &= CA^{(i)} h^{(1)}_{corrected} + y^{(2)}[i] \\
y^{(V)}_{corrected}[i] &= CA^{(i)} h^{(V-1)}_{corrected} + y^{(V)}[i]
\end{aligned}
\tag{4}
$$

Here $i \in [1, K]$ is the index of the output within the given subsequence. As shown, this allows us to compute the SSM in parallel across the devices, then correct the state afterwards, with a cheaper correction calculation.

**Training on even longer sequences.** Building on that, we further push the limits to train on sequence lengths of up to 15 million using gradient accumulation. Here, we split a very long sequence into subsequences that are as long as we can fit into device memory. We run the first subseqeunce and compute the loss, but rather than taking an optimization step, we save the gradients, and run on the next subsequence using the last state $h$ as the current state on the next subsequence. We can then repeat this process as many times as needed to reach any length sequence. We note however that this setting has a few drawbacks. First, it requires output and computing the loss for each subsequence, which means if for example its a question answering task, and the question hasn't been answered yet, it will result in some strange learning signal. Second, this introduces arbitrary boundaries in the learning, as the loss won't propagate over these boundaries. However, in practice we find these are not major limitations and the model is still able to learn from these long signals, and perform better than training without them (Table 1, 2).

## 3.2 ADDRESSING OVERFITTING

We find that training data and augmentation is especially important. We observed that the model very easily overfits when trained on encoded videos. However, if we train on individual frames encoded on JPEGs, we did not observe this overfitting. We realized that when training on individual frames we randomly sampled one of the frames, and as Kinetics videos are 10 seconds long, and we had data at 25 fps, we had roughly 250 frames we were sampling from, i.e., we roughly increased the data by 250 times. If instead we trained the model using only a single JPEG byte string per

| Model | PT Data | PT Modalities | Params | TFLOPS | K600 | K400 |
|---|---|---|---|---|---|---|
| ViViT-L (Arnab et al., 2021) | JFT-300M | Img | - | - | 82.9 | 83.5 |
| ViViT-H (Arnab et al., 2021) | JFT-300M | Img | - | - | 85.8 | 84.9 |
| MerlotReserve-H (Zellers et al., 2022) | YT-1B | Vid+Audio+Text | 644M | - | 91.1 | - |
| TubeViT-H (Piergiovanni et al., 2023) | ImageNet | Img | - | 17.64 | 91.8 | 90.9 |
| InternVideo2 (Wang et al., 2024b) | Many | Img+Vid+Audio+Text | 6B | - | 91.9 | 92.1 |
| VideoMamba (Li et al., 2024) | CLIP-400M | Img+Text | 74M | 28.42 | - | 85.0 |
| VideoMAE (Tong et al., 2022) | None | - | 600M | 88.76 | - | 87.4 |
| ST-MAE (Feichtenhofer et al., 2022) | IG-uncurated | Vid | 600M | 25.1 | - | 86.8 |
| Bytes-B (ours) | HowTo100M | Vid | 500M | 1.88 | 60.5 | 61.3 |
| Bytes-L (ours) | HowTo100M | Vid | 1B | 4.12 | 85.2 | 86.8 |

Table 3: Kinetics-600 and Kinetics-400 results. We note that our model is significantly cheaper than prior works, as shown in Table 13 and trained on much less data. PT stands for Pre-training.

video, e.g., the 100th frame, we saw the same overfitting behavior. In both these cases, when trained on a single encoded video or single encoded frame, the loss would go to 0 very quickly, while the evaluation accuracy would be extremely low, around 4%, regardless of input format. Based on this, we thought that the model would need significantly more training data, either in the form of data augmentation and/or pre-training.

Our next observation is that if we take two video clips and apply mild data augmentation, e.g., some slight color jittering, as is standard in training video models (e.g., ViViT (Arnab et al., 2021)), and encode them into mp4s, and compute the Levenshtein distance on the byte strings of these clips, we saw that over half the byte string is different. Visually, the two videos are indistinguishable. This suggests that the compressed bytes-based representations have a large amount of variation even with seemingly small changes to the visual inputs. Because of this, when training video byte based models, we apply a large amount of data augmentation during training. Specifically, we apply random temporal and spatial cropping, color jittering, contrast adjustment, color inversion, posterization, solarization, brightness, sharpness, and cutout augmentations. These augmentations are all done on the RGB space, then encoded into mp4s and used to train the model. We don't apply any augmentation to the bytes themselves (e.g., byte-level dropout, random substrings, etc), and leave explorations of that for future work.

**Self-supervised Pre-training.** We also explore self-supervised pre-training for video byte based models. Byte-based representations enable many different fully self-supervised tasks. First, we can train a model that takes the encoded video bytes as input and produces the RGB pixels of the video as output. However, as video generation is a complex task, itself having many specialized models and methods which are computationally intensive (e.g., video diffusion), and since we don't care about actually generating videos, just about training a video understanding model on byte-based representations, we simplify the RGB prediction significantly. Here, we take the output of the multilevel SSM model, and use an attention pooling layer to generate $F' \cdot H' \cdot W' \cdot D$ tokens. This is then reshaped to $F' \times H' \times W' \times D$, which gives us the rough size of the video. We then apply a small UNet-based (Ronneberger et al., 2015) model to upsample this to the video tensor $F \times H \times W \times 3$. We then apply a MSE loss between the predicted video and ground truth video RGB pixels. To further reduce compute, we generate only 10 frames per video (i.e., 1 fps for 10 second clips) at a low resolution of $128 \times 128$. While the reconstrustions do not look perfect, this provides a good enough learning signal to the model. Second, we explore a pre-training similar to how language models are trained: next byte prediction. I.e., the input to the model is a encoded mp4 byte string and the models task is to predict the next byte based on the previous bytes.

## 4 EXPERIMENTS

### 4.1 MAIN RESULTS

We present the main experimental results on the Kinetics-400 (Carreira & Zisserman, 2017), Kinetics-600 (Carreira et al., 2018), MLB YouTube (Piergiovanni & Ryoo, 2018), and Kinetics-Sounds (Arandjelovic & Zisserman, 2017) benchmarks, which are popular video or video+audio understanding benchmarks. Please see the appendix for implementation details (Sec. A.2).

| Model | | MAP |
|---|---|---|
| MLBYouTube | | 62.6 |
| Bytes-B | 500M | 63.5 |
| Bytes-L | 1B | 64.5 |

Table 4: MLBYouTube (Piergiovanni & Ryoo, 2018) results for fine-grained video understanding.

| Model | Audio | MAP |
|---|---|---|
| MBT (Nagrani et al., 2021) | yes | 85.0 |
| Bytes-L | no | 81.4 |
| Bytes-L | yes | 84.4 |

Table 5: Kinetics-Sounds (Arandjelovic & Zisserman, 2017) results. We see that including audio in the encoded video helps.

| Model | ActivityNet-QA | CinePile |
|---|---|---|
| VideoCoca (Yan et al., 2022) | 56.1 | |
| UMT-L (Li et al., 2023) | 47.9 | - |
| Mirasol-3B (Piergiovanni et al., 2024) | 51.1 | - |
| LLaVA-OV-7B | 56.6 | 49.3 |
| Bytes-L (1B) | 57.1 | 47.5 |

Table 6: Results on longer video understanding on AcitivityNet-QA and CinePile.

Table 3 shows the classification performance of the proposed method on the commonly used activity understanding benchmarks, Kinetics-400/Kinetics-600, with 400/600 classes. We note that Kinetics has 10 second videos, and we used 25 fps (far higher than previous works) giving us 250 frames per video, but after encoding, the input is only approximately 250,000 bytes long. We also note that prior works benefit from image-based pre-training, while we here only use video pre-training. Our results show strong performance, even when compared to larger video and video foundational models, despite using far less data and compute.

Table 4 further evaluates the performance on the MLB Youtube benchmark (Piergiovanni & Ryoo, 2018), which is a benchmark for distinguishing between fine-grained activities, which requires understanding motion at higher FPS than other datasets, like Kinetics. As seen, the model performs very well, outperforming the state-of-the-art approaches, using encoded bytes as input.

Our model easily extends to Audio+Video from bytes, showing competitive performance (Table 5).

Finally, we show results on longer videos (Table 6, see the appendix for details on this experiment).

## 4.2 MODEL EFFICIENCY

One advantage of the proposed approach is the compute and time savings since it operates on a highly compressed format. In Figure 4, we present the results comparing the FLOPs and runtime

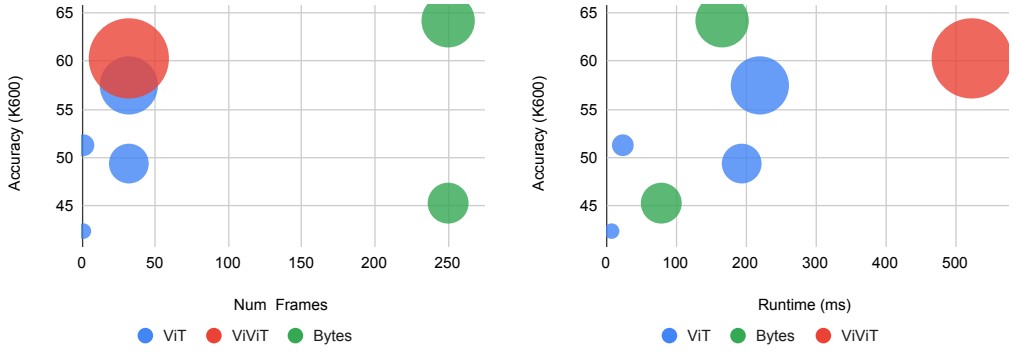

Figure 4: Plot of accuracy vs. runtime and frames, where the size of each model indicates how many FLOPs it uses. This shows the bytes models scale better to longer sequences, uses less FLOPs even as the model scales up, showing the potential of this approach.

| Model | K600 |
|---|---|
| 256 Embedding | 25.4 |
| 512 Embedding | 25.7 |
| 1024 Embedding | 21.2 |

Table 7: Scaling ablation on Kinetics, using Bytes-Tiny model. Increasing the embedding dimension of the model leads to overfitting.

| Method | TFLOPs | K600 |
|---|---|---|
| Attention Pooling | 1.33 | 23.9 |
| Average Pooling | 0.38 | 20.7 |
| Concatenation Pooling | 0.47 | 25.4 |

Table 8: Pooling method experiments, using Bytes-Tiny.

of the proposed model. As seen our models are much more lightweight. Furthermore, with other approaches a very limited number of frames can be sampled, e.g., up to 32, and these models become prohibitively expensive to sample more frames, whereas here the information content of all frames is processed which exceeds the upper bounds over number of frames for other models, e.g., all 250 frames on Kinetics (10 second clips at 25fps) for ours vs 32 or 64 for prior works.

**Parameter Efficiency Discussion** Encoded video bytes are a compressed representation and as such we found it took more parameters and training iterations to match the performance of pixel-based models. Models operating on pixels start with a representation that is already structured for (human) perception. Nearby pixels are spatially related, and basic patterns (edges, textures) are immediately available, and that is used by convolution/patch based models. However, starting from bytes, the model must first implicitly learn the 'language' of the codec. Thus some portion of the model's parameters are dedicated to solving this problem of 'decoding' the byte stream into a useful latent representation, which pixel-based models don't have to do, instead starting with the inductive bias of convolution/patches. In some sense, a bytes-based model is doing more with each parameter, since it has no inductive bias. We note that the FLOPs, runtime and memory usage of the model is significantly lower than pixel based ViTs, despite the larger parameter count. Since the relationship between FLOPs, runtime, memory usage, and parameters depends a lot on the network structure, the multilevel SSM on bytes-based data enables these compute savings even with more parameters.

### 4.3 ABLATIONS

For the ablations, we report the accuracy on Kinetics-600 using the Bytes-Tiny model.

**Model ablations**. Table 7 shows how the model scales with size, particularly, with increasing the size of the dimensionality of the feature representation in the model, which also results in larger model. As seen larger feature representation is beneficial, however, we observe overfitting for very large model sizes. Table 8 compares different versions of the pooling, we use concatenation pooling as it performs best for relatively small increase in FLOPs.

Table 9 explores several versions of the proposed model. As seen, using SSM-style models provides benefits for this application due to the sequence lengths and nature of the encoded byte structure, compared to a Transformer (Vaswani et al., 2017) model. The proposed multilevel SSM provides a further large jump in performance, as it has much higher capacity for increased sequence lengths. We also compare to a multilevel Transformer, which uses the same pooling methods as the multilevel SSM, but replaces the Mamba blocks with standard Transformer blocks.

We further compare to a causal convolution based model as well as the one proposed in (Horton et al., 2023), which uses many pooling layers and sliding window attention for encoded e.g., JPEG/PNG images. We note that here we are using longer sequences than those tested in that paper (262,144 vs. 150,000). For both, we used models matching the same parameter count as Bytes-Tiny. In Table 9 (Lines 3, 4), we find that our approach significantly outperforms these variants, as well.

**Pre-training Experiments**. Table 10 explores different methods for pre-training for the proposed models. We compare RGB reconstruction to next-byte prediction, finding that RGB reconstruction is slightly better as a pre-training task, but both are effective.

We also see how transferable byte-based models are. In Table 11, we compare no pre-training to a model pre-trained for RGB reconstruction vs a model pre-trained on JPEG bytes for classification. Previously, most video works used image pre-trained backbones which were then further trained on video data, and this experiment is similar to that. We see some benefit from pre-training with image

| Model | K600 |
|---|---|
| Transformer | N/A |
| Transformer with local attention with 512 tokens | 15.4 |
| Causal Conv (roughly equivalent params to our Bytes-Tiny) | 15.8 |
| BF-Ti Horton et al. (2023) | 17.5 |
| Multilevel Transformer | 16.7 |
| Mamba (Baseline SSM) | 18.4 |
| Bytes-Tiny (with Multilevel SSM) (ours) | 25.4 |

Table 9: Comparing different forms of the model. A standard Transformer with full global attention did not fit into memory with the long sequence lengths videos have (Bytes-Tiny model).

| Method | K600 |
|---|---|
| None | 25.4 |
| RGB prediction | 31.2 |
| Next-byte prediction | 28.6 |

Table 10: Effects of pre-training tasks, using the Bytes-Tiny model, on HowTo100M.

| Method | K600 |
|---|---|
| No Pre-training | 25.4 |
| JPEG Pre-trained | 27.2 |
| mp4 Pre-trained | 31.2 |

Table 11: Training on JPEG bytes and transferring to mp4 bytes, using Bytes-Tiny model.

| | K600 |
|---|---|
| 1x data | 2.4 |
| 10x data | 5.2 |
| 100x data | 18.6 |
| 200x data | 25.4 |

Table 12: Effects of data augmentation. Using Bytes-Tiny model.

JPEG bytes, compared to no pre-training, but it is not as good as video byte based pre-training. This is expected, but also shows these models are learning some generalization knowledge about encoded byte structures, even for very different encodings.

**Data Augmentation Effects**. Table 12 shows the effect of data augmentation when training. We generated a fixed number of samples by applying data augmentation as described above to generate 1 to 200 samples for each video. We find that increasing the data augmentation increases the performance of the model a lot.

**Preprocessing Cost Comparisons** We further compare the different preprocessing steps and times. In some cases, videos are stored in different formats, so a one-time transcoding operation may be needed. For these comparisons, we used the Kinetics videos. For the bytes input pipeline, we had a one-time transcoding cost of 193ms per video, 10ms to load the mp4 into memory, <1ms to transfer it to the TPU/GPU, for a total of 203ms. For pixel-based pipelines, we had 180ms to decode a mp4 into RGB pixels, 69ms for crop, resize, etc. 4ms to transfer 64 frames to TPU/GPU, a total of 254ms. These are very similar. The transcode, though, only needs to be done once, so for training, our input pipeline is significantly faster.

## 5 RELATED WORKS

Many works have studied efficient video representations, with some focused on compressed videos. For example (Wu et al., 2018) showed the benefit of using the compressed components (e.g., i-frames, p-frames) rather than decoding all frames. However, this work used a CNN on the pixels of the various frames formats. While this showed the potential, the use of the CNN and construction of the P-frames and i-frames removed some of the savings of working directly with the byte representation, and also designed a complex network to track motion over frames. Another work (Wiles et al., 2023) proposes learning neural codecs by learning a VQ-VAE model to compress videos. The model learns how to work in this compressed space, including things such as data augmentation, achieving good results on short video understanding (Kinetics) (Carreira & Zisserman, 2017). However, the bulk of the learning is placed on the neural codec model, while still using a CNN on top of the representation. Some works have explored learning from image bytes (Horton et al., 2023). However, the sequence lengths are only a few thousand bytes, limiting it to small images.

## 6 CONCLUSIONS

We propose a novel approach to understand videos from encoded and compressed byte representations. This has the advantage of saving memory and compute, compared to working on pixels, and better scales to longer sequences, reaching 15 Million. We show strong performance on this new and challenging task and demonstrate there is much potential in learning from raw video bytes.

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

## A  Supplemental Materials

### A.1  Pooling Methods

Given the input embedding sequence, $L \times D$, we input this to the SSM. Specifically, this model is the Mamba SSM architecture. We do not make any changes to the layers or operations themselves. To create the multilevel SSM, we instead add pooling or merging layers within the SSM. We explore several approaches to this: (1) using attention-based pooling layers, (2) average pooling, and (3) concatenation pooling.

**Attention Pooling.** Here, we pass a sub-sequence of length $L_s$ into the attention layer and output 1 token, thus pooling $L_s$ tokens. For example, given a sequence length of $S$, and $L_s = 4$, we reduce the sequence by 4 by creating a $\frac{S}{L_s} \times L_s \times D$ tensor, then applying attention pooling e.g., (Touvron et al., 2021). Specifically, we use a query vector with 1 latent embedding, when when applied to the input key/value tensor which has size $L_s \times D$, results in a $1 \times D$ output. When applied along the whole sequence, this results in a $\frac{S}{L_s} \times D$ tensor. This can then be passed to the next SSM block.

**Average pooling.** This is similar to the above approach, expect we apply average pooling over the $L_s$ dimension, resulting in the same $\frac{S}{L_s} \times D$ tensor, but without the attention operation, only averaging.

**Concatenation pooling.** Here, we create a tensor of shape $\frac{S}{L_s} \times D \cdot L_s$, by re-arranging the tensor to group multiple tokens into 1 by combining along the embedding axis. I.e., we reshape from $S \times D$ to $\frac{S}{L_s} \times D \cdot L_s$.

Finally, after any of the pooling layers, we apply a fully-connected layer to project the resulting tensor to the final dimension $D_{out}$, which can either be the same as $D$ or larger. We found $L_s = 4$ and $D_{out} = 2 \cdot D$ worked well in our experiments. This reduces the memory used by the sequence by a factor of 2 each time a pooling layer is applied.

### A.2  Implementation Details

We encode the videos as H.264 and in mp4 containers with a image size of 384 and a bitrate of 200kbps. This roughly matches resolution standard video models use. The model architectures used, Bytes-Tiny, Bytes-Base (Bytes-B), Bytes-Large (Bytes-L), are described in Table 16. Unless otherwise noted, we use only the video stream and do not include audio in the encoded video. We find the model is sensitive to learning rates, both different tasks (e.g., pre-training vs. classification finetuning) and model scales need different learning rates. We use $9e^{-5}$ as the pre-training learning rate for the Bytes-Tiny model, $7e^{-5}$ for the Bytes-Base model and $5e^{-5}$ for Bytes-Large. For fine-tuning, we use $5e^{-5}$, $3e^{-5}$, and $1e^{-5}$ for the tiny, base and large models, respectively. We pre-train with a batch size of 64, a sequence length of $2^18$ (262144) and for 2,000,000 steps. We fine-tune with the same settings, but for 1,000,000 steps. We note that with sufficient data augmentation, we do not observe overfitting behaviors even with 200 epochs of training on Kinetics-600. We use the Adam optimizer (Kingma, 2014), which is also important, with default settings. We use 512 TPU v5p to train the model. The Tiny model runs at approximately 20 steps per second, Base runs about 9 steps/sec and large about 4 steps/sec. Thus to train the model it takes about 27 hours, 61, and 127 hours to pre-train the models respectively. And about 14, 30, 63 hours for fine-tuning. In the Appendix (Table 16) we give the details for each model configuration used in the paper.

**Long Video Training** To reduce compute costs, we train the model in stages. First, we do the pre-training as above on short video segments. This provides us with a good base model that understands video bytes. Next, we do two stage of long video training, using the method described in Sec. 3.1. We train on the VideoMarathon (Lin et al., 2025) data in two stages, first with sequence lengths of 2 million then 15 million bytes, roughly 1.3 minutes and 10 minutes long. We emphasize here that we are training on the full video, without any subsampling, unlike prior works.

### A.3  Further Experiments

In Table 13 and Figure 5, we provide the rest of the efficiency comparisons. In Table 14, we compare the model using different codecs and containers. While these results show the performance is pretty

| Model | Num. Frames | Params | TFLOPs | Runtime | K600 |
|---|---|---|---|---|---|
| ViT-B | 1 frame | 86M | 0.05 | 8ms | 42.4 |
| ViT-B | 32 frame | 86M | 1.72 | 194ms | 49.4 |
| ViT-L | 1 frame | 307M | 0.16 | 24ms | 51.3 |
| ViT-L | 32 frames | 307M | 5.23 | 220ms | 57.5 |
| ViViT-L | 32 frames | >307M | 11.94 | 523ms | 60.3 |
| Bytes-B | all (250 frames) | 500M | 1.88 | 79ms | 45.3 |
| Bytes-L | all (250 frames) | 1B | 4.12 | 166ms | 64.2 |

Table 13: Model efficiency, we computed these values based on available implementations. ViTs are run per frame + average pooling before classification. Note that we did not use any pre-training for these models, just directly trained on Kinetics 600.

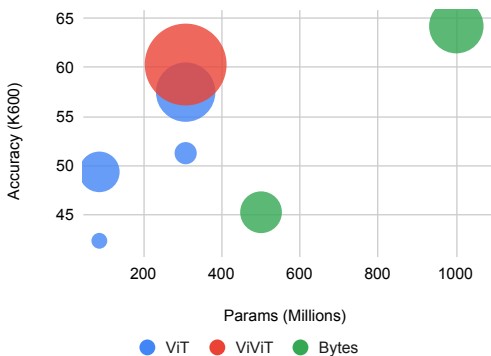

Figure 5: Plot of accuracy vs. params, where the size of each model indicates how many FLOPs it uses. Showing the bytes scale to larger models sizes while using fewer flops and gaining more accuracy.

similar across codecs, it is possible some are easier for the model to learn than others. In Table 15, we compare our proposed multilevel SSM to the chunked based one in (Bhirangi et al., 2024), the main difference being we apply the SSM to the entire sequence, while the other chunks the sequence and then applies the SSM independently to each chunk. Due to our efficient implementation of sequence parallelism, there is no meaningful difference in compute costs or runtime between the approaches, and for video byte inputs, applying the SSM to the whole sequence is better. In Table 16, we detail the configurations used for each model.

| Codec | Container | K600 |
|---|---|---|
| h.264 | mp4 | 25.4 |
| h.265 | mp4 | 25.1 |
| h.264 | mov | 25.8 |
| h.265 | mov | 24.7 |
| VP9 | mp4 | 24.3 |
| VP9 | WebM | 24.9 |

Table 14: Experiment showing performance of different video codecs and containers, using Bytes-Tiny. We see there is a small difference between the settings, but in general, they all perform very similarly.

| Model | K600 |
|---|---|
| Ours | 25.4 |
| Chunked (Bhirangi et al., 2024) | 25.1 |

Table 15: Multilevel SSM applied to the whole sequence vs. chunked (as in (Bhirangi et al., 2024). We note that there is no noticeable difference in FLOPs or compute time between these two approaches.

| **Model** | Layers | $D$ | $L_s$ | $N$ | $M$ | Params | TFLOPs |
|---|---|---|---|---|---|---|---|
| Bytes-Tiny | 12 | 256 | 4 | 4 | 3 | 103M | 0.47 |
| Bytes-B | 33 | 256 | 4 | 4 | 8 | 500M | 1.88 |
| Bytes-L | 45 | 256 | 4 | 4 | 11 | 1B | 4.12 |

Table 16: Model configs used in the paper.

## B  ACTIVITYNET-QA AND CINEPILE LONG VIDEO EXPERIMENTS

For the experiments in table 6, we needed to add a language model to handle the question answering task. To do this, we used the Gemma model Team et al. (2025a). Specifically, we then took the final representations from the SSM model as the video representation, and added that to the enmbedded text representations and then trained gemma to generate the answers.

For CinePile, since it is a multiple choice dataset, we evalute using standard accuracy, if the predicted answer (e.g., a, b, c, or d) matches the ground truth. For ActivityNet, since it is open-ended questions, we use string equality to compare the answers.

CinePile videos average 3 minutes of duration, with some as long as 8 minutes. It has both a training and evaluation set, so we finetune the 15M token model on this data for 1 epoch. We train the entire model with a learning rate of $0.00001$ on this question answering task.

ActivityNet-QA has videos with the average duration between 5 and 10 minutes. It also has a training and evaluation set, and we finetune for 1 epoch as well with a learning rate of $0.00001$.

## C  DISCUSSIONS ON SELF-SUPERVISED PRE-TRAINING

There are many other pre-training tasks could be explored, such as codec translation, e.g., mp4 (H.264) as input and generate a VP9 encoded video as output, using a standard per-token cross-entropy loss, or predicting features from a known visual encoder (e.g., CLIP (Radford et al., 2021) features) rather than directly predicting pixels. Similarly contrastive losses across different codecs could be used. Weakly supervised tasks such as predicted ASR transcripts from video byte inputs could be explored. We leave these explorations as future work.

