# OpenReview forum: "Learning from Encoded Video Bytes"
_ICLR.cc/2026/Conference — ICLR 2026 Conference Withdrawn Submission_

### Official Review · Reviewer_AS16 · 2025-10-29

**Soundness:** 2
**Presentation:** 3
**Contribution:** 3
**Rating:** 2
**Confidence:** 4

**Summary:**

This work propose to process video at the compressed byte level for video understanding. They especially designed a State-Space model for handling long byte sequences, and a multilevel SSM activation fusion for sequence length reduction and content understanding. They conducted experiments on popular video and video+audio benchmarks and compared with previous works.

**Strengths:**

1. Learning from compressed bytes for video understanding is a very novel and inspiring idea. Consuming videos without decoding into RGB frames has great potential of resource-saving, and this direction can be further explored by the community.

2. The work explored and contributed some study about the structure of the bytes and training knowledge of the bytes: data augmentation and pre-training are important for video byte model.

**Weaknesses:**

1. Table 3 is only indicating the TFLOPS benefits of bytes over RGB frames, but not showing the accuracy benefits / params benefits / latency benefits about that. Model Bytes obviously doesn't show strong classification performance, but which is claimed in lines 347-348.
Question: Do you have any comparisons with variable controlled? like, under the same pretraining setting / params level / TFLOPS level, if you intend to prove the accuracy benefits?

2. In lines 344-345, "we used 25 fps (far higher than previous works)", if you claim that, can you show the fps number at the table? How many fps is used by others?

3. In Table 4, what is the baseline model? what is indicated by the second column? Can you clearly show and compare the computation number and performance number in the table for a clear comparison and for reaching the conclusion? Same as in Table 5 (which is not surpassing baseline) and Table 6.

4. In Figure 4, I don't think 'number of frames - accuracy' constructs a proper trade-off pair in video understanding area. I also cannot see clear benefits to accuracy of sampling more frames. Num frames is not the number indicating the efficiency. Basically, accuracy should be compared under the same num frame setting. I never saw 'acc-num frame' is used as a trade-off pair to convince. Usually num frame can be used to indicate model throughput under the same computation/time resources. But it is weird to be set as x-axis in Figure 4.

5. In the right figure of Figure 4, I also cannot see obvious advantage of using compressed bytes over RGB frames, acc-runtime tradeoff.

**Questions:**

See weaknesses part.

---

### Official Review · Reviewer_ZNXf · 2025-10-29

**Soundness:** 1
**Presentation:** 1
**Contribution:** 2
**Rating:** 2
**Confidence:** 5

**Summary:**

This paper proposes, for the first time, training deep learning models on encoded video bytes as opposed to raw RGB frames. They claim that encoded video bytes are similar to RGB frames in the sense that they can be decoded to into RGB frames. Thus, the semantic mapping learned by deep models trained on RGB frames should also work with encoded bytes. Due to the increased sequence length of video bytes as opposed to RGB videos, they propose using state space models over transformers due to their linear complexity. They perform training on a large pre-training RGB video dataset with their "Bytes" model and show experiments on various downstream action recognition and long video understanding datasets.

**Strengths:**

1. The idea of training deep learning models with bytes vs. conventional RGB frames is extremely interesting.
2. Many experiments are provided to validate that training a deep learning model with video bytes is feasible.

**Weaknesses:**

1. **Extremely limited novelty** Besides the research problem of training deep learning models with encoded video bytes, there is no additional novelty introduced in the paper. The proposed Bytes model is basically a standard Mamba model, with stacked Mamba layers and some dimensional pooling in-between layers. The idea of computing sequences without a hidden state and updating after the fact was already proposed in the original Mamba paper [1]. The rest of the methodology section simply discuss trivial engineering details such as embedding size and pooling structure.

2. **Scattered/Incomplete Experiments** Many of the experiments and tables in the paper are incomplete or do not have a coherent message. In Table 1, why are multi-modal models used as a comparison? The main idea should be to compare simple, uni-modal models that were ideally pre-trained on the same dataset as the Bytes model for a fairer comparison. Instead, a random assortment of multi-modal models, pre-trained on all different datasets, are used as comparison. Furthermore, I question the validity of Table 3 as the only VideoMamba variant that has 28.4 TFLOPS was VideoMamba trained on K400 in a self-supervised manner, whereas the authors claim that the table denotes classification results.  Moreover, the fact that simply doubling the size of model leads to a 25% increase in performance is surprising, and yet there is no discussion whatsoever on that point. Further, Tables 4 and 5 show results on a relatively unknown dataset, with no meaningful baselines. These are just a few examples of the inconsistency/unclear narrative around the experiments (see below).

3. **Unsupported Claims** There are many claims made throughout the paper which require further support to confirm - especially given the new domain of training on encoded bytes. For example:

- "Bytes that are far apart have no correlation, so attention is not needed to understand video bytes." (Lines 127-128)
- "Recent SSM models [...] still don't fully address some challenges such as [...]. We present a different approach which preserves better the sequence signal." (Lines 160-165). The authors simply use stacked Mamba layers, so how does their method preserve the sequence better? Does this mean the comparison with baseline Mamba in Table 9 is a single Mamba layer?

- "We observed that the model very easily overfits when trained on encoded videos" (Line 265). Not only are none of these experiments shown, but does this also imply that without augmentations, the Bytes model would get 4% on any test data due to overfitting? Yet simply by applying augmentations to the RGB video, which in turn changes the byte string, the authors are then able to attain nearly 85% on Kinetics? This also further raises questions about if nearly "half the string" is changed when applying small augmentations in the RGB image, how are encoded bytes a reliable enough modality for deep learning models to perform video understanding? The fact that even image-level understanding is not tackled first before investigating video/long-video understanding leads me to believe there is a lot more experimentation required to validate the large claims made in this paper.

[1] Gu, A., & Dao, T. (2024, May). Mamba: Linear-time sequence modeling with selective state spaces. In First conference on language modeling.

**Questions:**

See weaknesses section. Overall, I think the research question is interesting, but the scope, narrative, experiments, and writing of the paper requires a lot of improvement.

---

### Official Review · Reviewer_qgTF · 2025-10-30

**Soundness:** 3
**Presentation:** 3
**Contribution:** 3
**Rating:** 6
**Confidence:** 2

**Summary:**

This paper presents a novel paradigm, learning video from encoded bytes. Compared to traditional RGB-based video models, the proposed State-Space model is capable of handling extra-long video tokens at training time. The model achieves competitive performance on video-only and video+audio understanding tasks.

**Strengths:**

1. Using video bytes as model input substantially reduces compute and storage, addressing a key bottleneck in scaling video models.

2. The proposed multi-stage SSM, parallelism strategy, strong data-augmentation are combined to facilitate the model training.

3. Compared to RGB-based video models, the proposed model yields competitive performance on stardard video understanding benchmarks.

**Weaknesses:**

1. Despite not perfect, the authors should give some cases on the reconstructed RGB/raw bytes in the self-supervised learning setup.

2. As data-augmentation is conducted in the RGB space, this raises a concern about efficiency. The decoding of video is usually time-consuming (e.g. via Decord), sometimes being the efficiency bottleneck of dataloader especially for long videos. It seems that the proposed decode-then-encode augmentation pipeline might also be affected.

3. It would be better if the results on SSv2 are reported, as SSv2 focuses on fine-grained temporal dynamics.

**Questions:**

1. As encoding video bytes shares some similar features with language modeling, is it possible to initialize the model (even partially) with LLM pre-trained weights?

---

### Official Review · Reviewer_jVJm · 2025-10-31

**Soundness:** 3
**Presentation:** 3
**Contribution:** 2
**Rating:** 4
**Confidence:** 4

**Summary:**

The paper proposes training video understanding models directly on compressed video byte streams instead of decoded RGB frames. It introduces a multilevel SSM to shorten sequences, and a sequence-parallel state correction scheme that enables training on very long inputs. The method achieves competitive results on multiple benchmarks.

The combination of compressed video input and state-space modeling is a promising design for scaling to long video sequences. However, the model design remains insufficiently developed, lacking clear mechanisms tailored to compressed data.

**Strengths:**

1, Clear motivation. The paper is well motivated. While most VLMs struggle to scale to long videos and rely on compressing image tokens from raw pixels, this work instead processes compressed video streams directly, leveraging mature video compression techniques to achieve efficient and scalable long-sequence modeling.

2, Enabling parallelism for SSMs. The model proposes a technique for parallelism for SSMs. The sequence-parallel + state-correction formulation (with explicit update equations) is effective and lets SSMs scale to 15M tokens in training.

3, Competitive results. The paper evaluates the performance in action recognition and  VQA benchmarks  and achieves competitive performance with strong efficiency scaling, including results on Kinetics-400/600 and long-video QA tasks (e.g., 57.1% on ActivityNet-QA with an LLM head).

**Weaknesses:**

1, Insufficient Related work. The Related Works section lacks prior research on video compression techniques (e.g., H.264, DCT, motion vectors), state-space models like S4 and Mamba, and compressed-domain learning from JPEG or H.264 inputs. Including these would better contextualize the proposed method and clarify its novelty.

2, Video Byte-based model design underexplored. The model design shows little difference from a standard image-input long-sequence model and lacks specific adaptations for encoded video byte inputs. While the paper focuses on the SSM architecture and long-sequence processing, it omits critical details on handling compressed video data, such as how temporal compression, quantization, and I/P/B frame dependencies are managed—especially since B-frames require both past and future frames while SSMs operate only on past states. It also remains unclear how performance would compare if the model used image-byte sequence as inputs instead of video bytes.

3, Parallelism assumptions vs. selective SSMs. The state-correction derivation (Eq. 1–4) is presented in linear SSM form; since Mamba is input-selective, it would help to clarify when/why these corrections remain valid and how approximation error behaves across chunks.


Minor:

1, In abstract, the abbreviation of SSM is used without introduction.

2, Texts in the figures are too small and hard to read.

**Questions:**

1, In line 240, which parameter is $h_0$ in equation 3? Would the correction still be valid when C is input dependent? How does the correction handle multilevel SSM with pooling?

2, How does the model encode the byte input? How does it handle I, B and P frame differently?

3, What is the model performance on long video benchmarks, such as LVBench, MLVU, VUE-TR or Neptune? (Line 73-74)

---

### Note · Authors · 2025-11-13

I have read and agree with the venue's withdrawal policy on behalf of myself and my co-authors.